# Maximum Entropy Fine-Grained Classification

**Abhimanyu Dubey  Otkrist Gupta  Ramesh Raskar  Nikhil Naik**
Massachusetts Institute of Technology
Cambridge, MA, USA
{dubeya, otkrist, raskar, naik}@mit.edu

## Abstract

Fine-Grained Visual Classification (FGVC) is an important computer vision problem that involves small diversity within the different classes, and often requires expert annotators to collect data. Utilizing this notion of small visual diversity, we revisit Maximum-Entropy learning in the context of fine-grained classification, and provide a training routine that maximizes the entropy of the output probability distribution for training convolutional neural networks on FGVC tasks. We provide a theoretical as well as empirical justification of our approach, and achieve state-of-the-art performance across a variety of classification tasks in FGVC, that can potentially be extended to any fine-tuning task. Our method is robust to different hyperparameter values, amount of training data and amount of training label noise and can hence be a valuable tool in many similar problems.

## 1  Introduction

For ImageNet [7] classification and similar large-scale classification tasks that span numerous diverse classes and millions of images, strongly discriminative learning by minimizing the cross-entropy from the labels improves performance for convolutional neural networks (CNNs). Fine-grained visual classification problems differ from such large-scale classification in two ways: (i) the classes are visually very similar to each other and are harder to distinguish between (see Figure 1a), and (ii) there are fewer training samples and therefore the training dataset might not be representative of the application scenario. Consider a technique that penalizes strongly discriminative learning, by preventing a CNN from learning a model that memorizes specific artifacts present in training images in order to minimize the cross-entropy loss from the training set. This is helpful in fine-grained classification: for instance, if a certain species of bird is mostly photographed against a different background compared to other species, memorizing the background will lower generalization performance while lowering training cross-entropy error, since the CNN will associate the background to the bird itself.

In this paper, we formalize this intuition and revisit the classical *Maximum-Entropy* regime, based on the following underlying idea: the entropy of the probability logit vector produced by the CNN is a measure of the "peakiness" or "confidence" of the CNN. Learning CNN models that have a higher value of output entropy will reduce the "confidence" of the classifier, leading in better generalization abilities when training with limited, fine-grained training data. Our contributions can be listed as follows: (i) we formalize the notion of "fine-grained" vs "large-scale" image classification based on a measure of diversity of the features, (ii) we derive bounds on the $\ell_2$ regularization of classifier weights based on this diversity and entropy of the classifier, (iii) we provide uniform convergence bounds on estimating entropy from samples in terms of feature diversity, (iv) we formulate a fine-tuning objective function that obtains state-of-the-art performance on five most-commonly used FGVC datasets across six widely-used CNN architectures, and (v) we analyze the effect of Maximum-Entropy training over different hyperparameter values, amount of training data, and amount of training label noise to demonstrate that our method is consistently robust to all the above.

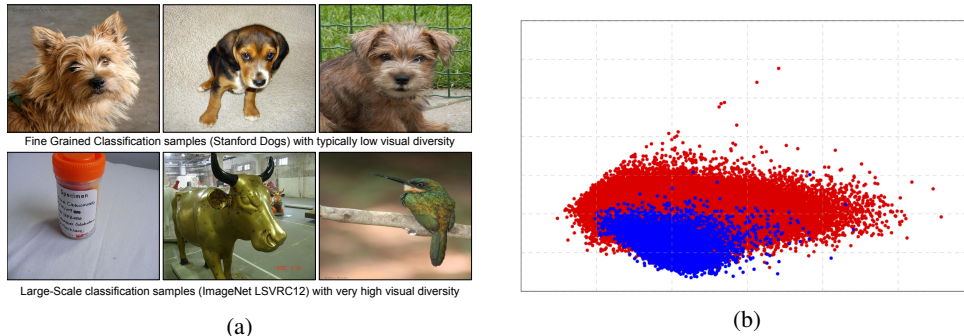

Fine Grained Classification samples (Stanford Dogs) with typically low visual diversity

Large-Scale classification samples (ImageNet LSVRC12) with very high visual diversity

(a)　　　　　　　　　　　　　　　　　　　(b)

Figure 1: (a) Samples from the CUB-200-2011 FGVC (top) and ImageNet (bottom) datasets. (b) Plot of top 2 principal components (obtained from ILSVRC-training set on GoogleNet `pool5` features) on ImageNet (red) and CUB-200-2011 (blue) validation sets. CUB-200-2011 data is concentrated with less diversity, as hypothesized.

## 2 Related Work

**Maximum-Entropy Learning:** The principle of Maximum-Entropy, proposed by Jaynes [16] is a classic idea in Bayesian statistics, and states that the probability distribution best representing the current state of knowledge is the one with the largest entropy, in context of testable information (such as accuracy). This idea has been explored in different domains of science, from statistical mechanics [1] and Bayesian inference [12] to unsupervised learning [8] and reinforcement learning [29, 27]. Regularization methods that penalize minimum entropy predictions have been explored in the context of semi-supervised learning [11], and on deterministic entropy annealing [36] for vector quantization. In the domain of machine learning, the regularization of the entropy of classifier weights has been used empirically [4, 42] and studied theoretically [37, 49].

In most treatments of the Maximum-Entropy principle in classification, emphasis has been given to the entropy of the weights of classifiers themselves [37]. In our formulation, we focus instead on the Maximum-Entropy principle applied to the prediction vectors. This formulation has been explored experimentally in the work of Pereyra et al.[33] for generic image classification. Our work builds on their analysis by providing a theoretical treatment of fine-grained classification problems, and justifies the application of Maximum-Entropy to target scenarios with limited diversity between classes with limited training data. Additionally, we obtain large improvements in fine-grained classification, which motivates the usage of the Maximum-Entropy training principle in the fine-tuning setting, opening up this idea to much broader range of applied computer vision problems. We also note the related idea of *label smoothing* regularization [41], which tries to prevent the largest logit from becoming much larger than the rest and shows improved generalization in large scale image classification problems.

**Fine-Grained Classification:** Fine-Grained Visual Classification (FGVC) has been an active area of interest in the computer vision community. Typical fine-grained problems such as differentiating between animal and plant species, or types of food. Since background context can act as a distraction in most cases of FGVC, there has been research in improving the attentional and localization capabilities of CNN-based algorithms. Bilinear pooling [25] is an instrumental method that combines pairwise local features to improve spatial invariance. This has been extended by Kernel Pooling [6] that uses higher-order interactions instead of dot products proposed originally, and Compact Bilinear Pooling [9] that speeds up the bilinear pooling operation. Another approach to localization is the prediction of an affine transformation of the original image, as proposed by Spatial Transformer Networks [15]. Part-based Region CNNs [35] use region-wise attention to improve local features. Leveraging additional information such as pose and regions have also been explored [3, 46], along with robust image representations such as CNN filter banks [5], VLAD [17] and Fisher vectors [34]. Supplementing training data [21] and model averaging [30] have also had significant improvements.

The central theme among current approaches is to increase the diversity of relevant features that are used in classification, either by removing irrelevant information (such as background) by better localization or pooling, or supplementing features with part and pose information, or more training data. Our method focuses on the classification task after obtaining features (and is hence compatible with existing approaches), by selecting the classifier that assumes the minimum information about the task by principle of Maximum-Entropy. This approach is very useful in context of fine-grained tasks, especially when fine-tuning from ImageNet CNN models that are already over-parameterized.

# 3 Method

In the case of Maximum Entropy fine-tuning, we optimize the following objective:

$$\boldsymbol{\theta}^* = \arg\min_{\boldsymbol{\theta}} \widehat{\mathbb{E}}_{\mathbf{x} \sim \mathcal{D}} \left[ \mathbb{D}_{\mathsf{KL}} \left( \bar{\mathbf{y}}(\mathbf{x}) || p(\mathbf{y}|\mathbf{x}; \boldsymbol{\theta}) \right) - \gamma \mathsf{H}[p(\mathbf{y}|\mathbf{x}; \boldsymbol{\theta})] \right] \tag{1}$$

Where $\boldsymbol{\theta}$ represents the model parameters, and is initialized using a pretrained model such as ImageNet [7] and $\gamma$ is a hyperparameter. The entropy can be understood as a measure of the "peakiness" or "indecisiveness" of the classifier in its prediction for the given input. For instance, if the classifier is strongly confident in its belief of a particular class $k$, then all the mass will be concentrated at class $k$, giving us an entropy of 0. Conversely, if a classifier is equally confused between all $C$ classes, we will obtain a value of $\log(C)$ of the entropy, which is the maximum value it can take. In problems such as fine-grained classification, where samples that belong to different classes can be visually very similar, it is a reasonable idea to prevent the classifier from being too confident in its outputs (have low entropy), since the classes themselves are so similar.

## 3.1 Preliminaries

Consider the multi-class classification problem over $C$ classes. The input domain is given by $\mathcal{X} \subset \mathbb{R}^Z$, with an accompanying probability metric $p_{\mathsf{x}}(\cdot)$ defined over $\mathcal{X}$. The training data is given by $N$ i.i.d. samples $\mathcal{D} = \{\mathbf{x}_1, ..., \mathbf{x}_N\}$ drawn from $\mathcal{X}$. Each point $\mathbf{x} \in \mathcal{X}$ has an associated label $\bar{\mathbf{y}}(\mathbf{x}) = [0, ..., 1, ...0] \in \mathbb{R}^C$. We learn a CNN such that for each point in $\mathcal{X}$, the CNN induces a conditional probability distribution over the $m$ classes whose mode matches the label $\bar{\mathbf{y}}(\mathbf{x})$.

A CNN architecture consists of a series of convolutional and subsampling layers that culminate in an activation $\Phi(\cdot)$, which is fed to an $C$-way classifier with weights $\mathbf{w} = \{\mathbf{w}_1, ..., \mathbf{w}_C\}$ such that:

$$p(y_i|\mathbf{x}; \mathbf{w}, \Phi(\cdot)) = \frac{\exp\left(\mathbf{w}_i^\top \Phi(\mathbf{x})\right)}{\sum_{j=1}^{C} \exp\left(\mathbf{w}_j^\top \Phi(\mathbf{x})\right)} \tag{2}$$

During training, we learn parameters $\mathbf{w}$ and feature extractor $\Phi(\cdot)$ (collectively referred to as $\boldsymbol{\theta}$), by minimizing the expected KL (Kullback-Liebler)-divergence of the CNN conditional probability distribution from the true label vector over the training set $\mathcal{D}$:

$$\boldsymbol{\theta}^* = \arg\min_{\boldsymbol{\theta}} \widehat{\mathbb{E}}_{\mathbf{x} \sim \mathcal{D}} \left[ \mathbb{D}_{\mathsf{KL}} \left( \bar{\mathbf{y}}(\mathbf{x}) || p(\mathbf{y}|\mathbf{x}; \boldsymbol{\theta}) \right) \right] \tag{3}$$

During fine-tuning, we learn a feature map $\Phi(\cdot)$ from a large training set (such as ImageNet), discard the original classifier $\mathbf{w}$ (referred now onwards as $\mathbf{w}_S$) and learn new weights $\mathbf{w}$ on the smaller dataset (note that the number of classes, and hence the shape of $\mathbf{w}$, may also change for the new task). The entropy of conditional probability distribution in Equation 2 is given by:

$$\mathsf{H}[p(\cdot|\mathbf{x}; \boldsymbol{\theta})] \triangleq - \sum_{i=1}^{m} p(y_i|\mathbf{x}; \boldsymbol{\theta}) \log(p(y_i|\mathbf{x}; \boldsymbol{\theta})) \tag{4}$$

To minimize the overall entropy of the classifier over a data distribution $\mathbf{x} \sim p_{\mathsf{x}}(\cdot)$, we would be interested in the expected value of the entropy over the distribution:

$$\mathbb{E}_{\mathsf{x} \sim p_{\mathsf{x}}} \left[ \mathsf{H}[p(\cdot|\mathbf{x}; \boldsymbol{\theta})] \right] = \int_{\mathbf{x} \sim p_{\mathsf{x}}} \mathsf{H}[p(\cdot|\mathbf{x}; \boldsymbol{\theta})] p_{\mathsf{x}}(\mathbf{x}) d\mathbf{x} \tag{5}$$

Similarly, the empirical average of the conditional entropy over the training set $\mathcal{D}$ is:

$$\widehat{\mathbb{E}}_{\mathbf{x} \sim \mathcal{D}}[\mathsf{H}[p(\cdot|\mathbf{x}; \boldsymbol{\theta})]] = \frac{1}{N} \sum_{i=1}^{N} \mathsf{H}[p(\cdot|\mathbf{x}_i; \boldsymbol{\theta})] \tag{6}$$

To have high training accuracy, we do not need to learn a model that gives zero cross-entropy loss. Instead, we only require a classifier to output a conditional probability distribution whose $\arg\max$ coincides with the correct class. Next, we show that for problems with low diversity, higher validation accuracy can be obtained with a higher entropy (and higher training cross-entropy). We now formalize the notion of diversity in feature vectors over a data distribution.

## 3.2 Diversity and Fine-Grained Visual Classification

We assume the pretrained $n$-dimensional feature map $\Phi(\cdot)$ to be a multivariate mixture of $m$ Gaussians, where $m$ is unknown (and may be very large). Using an overall mean subtraction, we can re-center the Gaussian distribution to be zero-mean. $\Phi(\mathbf{x})$ for $\mathbf{x} \sim p_\mathsf{x}$ is then given by:

$$\Phi(\mathbf{x}) \sim \sum_{i=1}^{m} \alpha_i \mathcal{N}(\boldsymbol{\mu}_i, \boldsymbol{\Sigma}_i), \text{ where } \mathbf{x} \sim p_\mathsf{x}, \alpha_i > 0 \ \forall i \text{ and } \mathbb{E}_{\mathbf{x} \sim p_\mathsf{x}}[\Phi(\mathbf{x})] = 0, \tag{7}$$

where $\boldsymbol{\Sigma}_i$s are $n$-dimensional covariance matrices for each class $i$, and $\boldsymbol{\mu}_i$ is the mean feature vector for class $i$. The zero-mean implies that $\bar{\boldsymbol{\mu}} = \sum_{i=1}^{m} \alpha_i \boldsymbol{\mu}_i = \mathbf{0}$. For this distribution, the equivalent covariance matrix can be given by:

$$\mathsf{Var}[\Phi(\mathbf{x})] = \sum_{i=1}^{m} \alpha_i \boldsymbol{\Sigma}_i + \sum_{i=1}^{m} \alpha_i (\boldsymbol{\mu}_i - \bar{\boldsymbol{\mu}})(\boldsymbol{\mu}_i - \bar{\boldsymbol{\mu}})^\top = \sum_{i=1}^{m} \alpha_i (\boldsymbol{\Sigma}_i + \boldsymbol{\mu}_i \boldsymbol{\mu}_i^\top) \triangleq \boldsymbol{\Sigma}^* \tag{8}$$

Now, the eigenvalues $\lambda_1, ..., \lambda_n$ of the overall covariance matrix $\boldsymbol{\Sigma}^*$ characterize the variance of the distribution across $n$ dimensions. Since $\boldsymbol{\Sigma}^*$ is positive-definite, all eigenvalues are positive (this can be shown using the fact that each covariance matrix is itself positive-definite, and $\mathsf{diag}(\boldsymbol{\mu}_i \boldsymbol{\mu}_i^\top)_k = (\mu_i^k)^2 \geq 0 \ \forall i, k$). Thus, to describe the variance of the feature distribution we define *Diversity*.

**Definition 1.** *Let the data distribution be $p_\mathsf{x}$ over space $\mathcal{X}$, and feature extractor be given by $\Phi(\cdot)$. Then, the Diversity $\boldsymbol{\nu}$ of the features is defined as:*

$$\boldsymbol{\nu}(\Phi, p_\mathsf{x}) \triangleq \sum_{i=1}^{n} \lambda_i, \ \text{ where } \{\lambda_1, ..., \lambda_n\} \text{ satisfy } \det(\boldsymbol{\Sigma}^* - \lambda_i \mathbf{I}_n) = 0$$

This definition of diversity is consistent with multivariate analysis, and is a common measure of the total variance of a data distribution [18]. Now, let $p_\mathsf{x}^L(\cdot)$ denote the data distribution under a large-scale image classification task such as ImageNet, and let $p_\mathsf{x}^F(\cdot)$ denote the data distribution under a fine-grained image classification task. We can then characterize fine-grained problems as data distributions $p_\mathsf{x}^F(\cdot)$ for any feature extractor $\Phi(\cdot)$ that have the property:

$$\boldsymbol{\nu}(\Phi, p_\mathsf{x}^F) \ll \boldsymbol{\nu}(\Phi, p_\mathsf{x}^L) \tag{9}$$

On plotting pretrained $\Phi(\cdot)$ for both the ImageNet validation set and the validation set of CUB-200-2011 (a fine-grained dataset), we see that the CUB-200-2011 features are concentrated with a lower variance compared to the ImageNet training set (see Figure 1b), consistent with Equation 9. In the next section, we describe the connections of Maximum-Entropy with model selection in fine-grained classification.

## 3.3 Maximum-Entropy and Model Selection

By the Tikhonov regularization of a linear classifier [10], we would want to select $\mathbf{w}$ such that $\sum_j \|\mathbf{w}_j\|_2^2$ is small ($\ell_2$ regularization), to get higher generalization performance. This technique is also implemented in neural networks trained using stochastic gradient descent (SGD) by the process of "weight-decay". Several recent works around obtaining spectrally-normalized risk bounds for neural networks have demonstrated that the excess risk scales with the Frobenius norm of the weights [31, 2]. Our next result provides some insight into how fine-grained problems can potentially limit model selection, by analysing the *best-case* generalization gap (difference between training and expected risk). We use the following result to lower-bound the norm of the weights $\|\mathbf{w}\|_2 = \sqrt{\sum_{i=1}^{C} \|\mathbf{w}_i\|_2^2}$ in terms of the expected entropy and the feature diversity:

**Theorem 1.** *Let the final layer weights be denoted by $\mathbf{w} = \{\mathbf{w}_1, ..., \mathbf{w}_C\}$, the data distribution be $p_\mathsf{x}$ over $\mathcal{X}$, and feature extractor be given by $\Phi(\cdot)$. For the expected condtional entropy, the following holds true:*

$$\|\mathbf{w}\|_2 \geq \frac{\log(C) - \mathbb{E}_{\mathbf{x} \sim p_\mathsf{x}}[\mathsf{H}[p(\cdot|\mathbf{x}; \boldsymbol{\theta})]]}{2\sqrt{\boldsymbol{\nu}(\Phi, p_\mathsf{x})}}$$

A full proof of Theorem 1 is included in the supplement. Let us consider the case when $\boldsymbol{\nu}(\Phi, p_\mathsf{x})$ is large (ImageNet classification). In this case, this lower bound is very weak and inconsequential. However, in the case of small $\boldsymbol{\nu}(\Phi, p_\mathsf{x})$ (fine-grained classification), the denominator is small, and this lower bound can subsequently limit the space of model selection, by only allowing models with large values of weights, leading to a larger *best-case* generalization gap (that is, when, Theorem 1 holds with equality). We see that if the numerator is small, the diversity of the features has a smaller impact on limiting the model selection, and hence, it can be advantageous to maximize prediction entropy. We note that since this is a lower bound, the proof is primarily expository and we can only comment on *best-case* generalization performance.

More intuitively, however, it can be understood that problems that are fine-grained will often require more information to distinguish between classes, and regularizing the prediction entropy prevents creating models that memorize a lot of information about the training data, and thus can potentially benefit generalization. In this sense, using a Maximum-Entropy objective function is similar to an online *calibration* of neural network predictions [13], to account for fine-grained problems. Now, Theorem 1 involves the expected conditional entropy over the data distribution. However, during training we only have sample access to the data distribution, which we can use as a surrogate. It is essential to then ensure that the empirical estimate of the conditional entropy (from $N$ training samples) is an accurate estimate of the true expected conditional entropy. The next result ensures that for large $N$, in a fine-grained classification problem, the sample estimate of average conditional entropy is close to the expected conditional entropy.

**Theorem 2.** *Let the final layer weights be denoted by* $\mathbf{w} = \{\mathbf{w}_1, ..., \mathbf{w}_C\}$, *the data distribution be* $p_\mathsf{x}$ *over* $\mathcal{X}$, *and feature extractor be given by* $\Phi(\cdot)$. *With probability at least* $1 - \delta > \frac{1}{2}$ *and* $\|\mathbf{w}\|_\infty = \max(\|\mathbf{w}_1\|_2, ..., \|\mathbf{w}_C\|_2)$, *we have:*

$$\left| \widehat{\mathbb{E}}_\mathcal{D}[\mathsf{H}[p(\cdot|\mathbf{x}; \boldsymbol{\theta})]] - \mathbb{E}_{\mathbf{x} \sim p_\mathsf{x}}[\mathsf{H}[p(\cdot|\mathbf{x}; \boldsymbol{\theta})]] \right| \leq \|\mathbf{w}\|_\infty \left( \sqrt{\frac{2}{N} \boldsymbol{\nu}(\Phi, p_\mathsf{x}) \log(\frac{4}{\delta})} + \widetilde{\mathcal{O}}(N^{-0.75}) \right)$$

A full proof of Theorem 2 is included in the supplement. We see that as long as the diversity of features is small, and $N$ is large, our estimate for entropy will be close to the expected value. Using this result, we can express Theorem 1 in terms of the empirical mean conditional entropy.

**Corollary 1.** *With probability at least* $1 - \delta > \frac{1}{2}$, *the empirical mean conditional entropy follows:*

$$\|\mathbf{w}\|_2 \geq \frac{\log(C) - \widehat{\mathbb{E}}_{\mathbf{x} \sim \mathcal{D}}[\mathsf{H}[p(\cdot|\mathbf{x}; \boldsymbol{\theta})]]}{\left( 2 - \sqrt{\frac{2}{N} \log(\frac{2}{\delta})} \right) \sqrt{\boldsymbol{\nu}(\Phi, p_\mathsf{x})} - \widetilde{\mathcal{O}}(N^{-0.75})}$$

A full proof of Corollary 1 is included in the supplement. We see that we recover the result from Theorem 1 as $N \to \infty$. Corollary 1 shows that as long as the diversity of features is small, and $N$ is large, the same conclusions drawn from Theorem 1 apply in the case of the empirical mean entropy as well. We will now proceed to describing the results obtained from maximum-entropy fine-grained classification.

# 4 Experiments

We perform all experiments using the PyTorch [32] framework over a cluster of NVIDIA Titan X GPUs. We now describe our results on benchmark datasets in fine-grained recognition and some ablation studies.

## 4.1 Fine-Grained Visual Classification

Maximum-Entropy training improves performance across five standard fine-grained datasets, with substantial gains in low-performing models. We obtain state-of-the-art results on all five datasets (Table 1-(A-E)). Since all these datasets are small, we report numbers averaged over 6 trials.

**Classification Accuracy:** First, we observe that Maximum-Entropy training obtains significant performance gains when fine-tuning from models trained on the ImageNet dataset (e.g., GoogLeNet

| (A) CUB-200-2011 [44] | | |
| --- | --- | --- |
| Method | Top-1 | Δ |
| *Prior Work* | | |
| STN[15] | 84.10 | - |
| Zhang et al. [47] | 84.50 | - |
| Lin et al. [24] | 85.80 | - |
| Cui et al. [6] | 86.20 | - |
| *Our Results* | | |
| GoogLeNet | 68.19 | (**6.18**) |
| **MaxEnt**-GoogLeNet | 74.37 | |
| ResNet-50 | 75.15 | (5.22) |
| **MaxEnt**-ResNet-50 | 80.37 | |
| VGGNet16 | 73.28 | (3.74) |
| **MaxEnt**-VGGNet16 | 77.02 | |
| Bilinear CNN [25] | 84.10 | (1.17) |
| **MaxEnt**-BilinearCNN | 85.27 | |
| DenseNet-161 | 84.21 | (2.33) |
| **MaxEnt**-DenseNet-161 | **86.54** | |

| (B) Cars [22] | | |
| --- | --- | --- |
| Method | Top-1 | Δ |
| *Prior Work* | | |
| Wang et al. [45] | 85.70 | - |
| Liu et al. [26] | 86.80 | - |
| Lin et al. [24] | 92.00 | - |
| Cui et al. [6] | 92.40 | - |
| *Our Results* | | |
| GoogLeNet | 84.85 | (2.17) |
| **MaxEnt**-GoogLeNet | 87.02 | |
| ResNet-50 | 91.52 | (2.33) |
| **MaxEnt**-ResNet-50 | **93.85** | |
| VGGNet16 | 80.60 | (3.28) |
| **MaxEnt**-VGGNet16 | 83.88 | |
| Bilinear CNN [25] | 91.20 | (1.61) |
| **MaxEnt**-Bilinear CNN | 92.81 | |
| DenseNet-161 | 91.83 | (1.18) |
| **MaxEnt**-DenseNet-161 | 93.01 | |

| (C) Aircrafts [28] | | |
| --- | --- | --- |
| Method | Top-1 | Δ |
| *Prior Work* | | |
| Simon et al. [38] | 85.50 | - |
| Cui et al. [6] | 86.90 | - |
| LRBP [20] | 87.30 | - |
| Lin et al. [24] | 88.50 | - |
| *Our Results* | | |
| GoogLeNet | 74.04 | (**5.12**) |
| **MaxEnt**-GoogLeNet | 79.16 | |
| ResNet-50 | 81.19 | (2.67) |
| **MaxEnt**-ResNet-50 | 83.86 | |
| VGGNet16 | 74.17 | (3.91) |
| **MaxEnt**-VGGNet16 | 78.08 | |
| BilinearCNN [25] | 84.10 | (2.02) |
| **MaxEnt**-BilinearCNN | 86.12 | |
| DenseNet-161 | 86.30 | (3.46) |
| **MaxEnt**-DenseNet-161 | **89.76** | |

| (D) NABirds [43] | | |
| --- | --- | --- |
| Method | Top-1 | Δ |
| *Prior Work* | | |
| Branson et al. [3] | 35.70 | - |
| Van et al. [43] | 75.00 | - |
| *Our Results* | | |
| GoogLeNet | 70.66 | (2.38) |
| **MaxEnt**-GoogLeNet | 73.04 | |
| ResNet-50 | 63.55 | (**5.66**) |
| **MaxEnt**-ResNet-50 | 69.21 | |
| VGGNet16 | 68.34 | (4.28) |
| **MaxEnt**-VGGNet16 | 72.62 | |
| BilinearCNN [25] | 80.90 | (1.76) |
| **MaxEnt**-BilinearCNN | 82.66 | |
| DenseNet-161 | 79.35 | (3.67) |
| **MaxEnt**-DenseNet-161 | **83.02** | |

| (E) Stanford Dogs [19] | | |
| --- | --- | --- |
| Method | Top-1 | Δ |
| *Prior Work* | | |
| Zhang et al. [48] | 80.43 | - |
| Krause et al. [21] | 80.60 | - |
| *Our Results* | | |
| GoogLeNet | 55.76 | (**6.25**) |
| **MaxEnt**-GoogLeNet | 62.01 | |
| ResNet-50 | 69.92 | (3.64) |
| **MaxEnt**-ResNet-50 | 73.56 | |
| VGGNet16 | 61.92 | (3.52) |
| **MaxEnt**-VGGNet16 | 65.44 | |
| BilinearCNN [25] | 82.13 | (1.05) |
| **MaxEnt**-BilinearCNN | 83.18 | |
| DenseNet-161 | 81.18 | (2.45) |
| **MaxEnt**-DenseNet-161 | **83.63** | |

Table 1: Maximum-Entropy training (**MaxEnt**) obtains state-of-the-art performance on five widely-used fine-grained visual classification datasets (A-E). Improvement over the baseline model is reported as (Δ). All results averaged over 6 trials.

[40], Resnet-50 [14]). For example, on the CUB-200-2011 dataset, fine-tuning GoogLeNet by standard fine-tuning gives an accuracy of 68.19%. Fine-tuning with Maximum-Entropy gives an accuracy of **74.37**%—which is a large improvement, and it is persistent across datasets. Since a lot of fine-tuning tasks use general base models such as GoogLeNet and ResNet, this result is relevant to the large number of applications that involve fine-tuning on specialized datasets.

Maximum-Entropy classification also improves prediction performance for CNN architectures specifically designed for fine-grained visual classification. For instance, it improves the performance of the Bilinear CNN [25] on all 5 datasets and obtains state-of-the-art results, to the best of our knowledge. The gains are smaller, since these architectures improve diversity in the features by localization, and hence maximizing entropy is less crucial in this case. However, it is important to note that most pooling architectures [25] use a large model as a base-model (such as VGGNet [39]) and have an expensive pooling operation. Thus they are computationally very expensive, and infeasible for tasks that have resource constraints in terms of data and computation time.

**Increase in Generality of Features:** We hypothesize that Maximum-Entropy training will encourage the classifier to reduce the specificity of the features. To evaluate this hypothesis, we perform the eigendecomposition of the covariance matrix on the `pool5` layer features of GoogLeNet trained on CUB-200-2011, and analyze the trend of sorted eigenvalues (Figure 2a). We examine the features from CNNs with (i) no fine-tuning ("Basic"), (ii) regular fine-tuning, and (iii) fine-tuning with Maximum-Entropy.

For a feature matrix with large covariance between the features of different classes, we would expect the first few eigenvalues to be large, and the rest to diminish quickly, since fewer orthogonal components can summarize the data. Conversely, in a completely uncorrelated feature matrix, we would see a longer tail in the decreasing magnitudes of eigenvalues. Figure 2a shows that for the Basic features (with no fine-tuning), there is a fat tail in both training and test sets due to the presence of a large number of uncorrelated features. After fine-tuning on the training data, we observe a

| Method | CIFAR-10 | $\Delta$ | CIFAR-100 | $\Delta$ |
|---|---|---|---|---|
| GoogLeNet | 84.16 | | 70.24 | |
| **MaxEnt** + GoogLeNet | 84.10 | (-0.06) | **73.50** | (**3.26**) |
| DenseNet-121 | 92.19 | | 75.01 | |
| **MaxEnt** + DenseNet-121 | 92.22 | (0.03) | **76.22** | (**1.21**) |

Table 2: Maximum Entropy obtains larger gains on the finer CIFAR-100 dataset as compared to CIFAR-10. Improvement over the baseline model is reported as ($\Delta$).

| Method | Random-ImageNet | $\Delta$ | Dogs-ImageNet | $\Delta$ |
|---|---|---|---|---|
| GoogLeNet | 71.85 | | 62.28 | |
| **MaxEnt** + GoogLeNet | 72.20 | (0.35) | 64.91 | (**2.63**) |
| ResNet-50 | 82.01 | | 73.81 | |
| **MaxEnt** + ResNet-50 | 82.29 | (0.28) | 75.66 | (**1.86**) |

Table 3: Maximum Entropy obtains larger gains on the a subset of ImageNet containing dog sub-classes versus a randomly chosen subset of the same size which has higher visual diversity. Improvement over the baseline model (in cross-validation) is reported as ($\Delta$).

reduction in the tail of the curve, implying that some generality in features has been introduced in the model through the fine-tuning. The test curve follows a similar decrease, justifying the increase in test accuracy. Finally, for Maximum-Entropy, we observe a substantial decrease in the width of the tail of eigenvalue magnitudes, suggesting a larger increase in generality of features in both training and test sets, which confirms our hypothesis.

**Effect on Prediction Probabilities:** For Maximum-Entropy training, the predicted logit vector is smoother, leading to a higher cross entropy during both training and validation. We observe that the average value of the logit probability of the top predicted class decreases significantly with Maximum-Entropy, as predicted by the mathematical formulation (for $\gamma = 1$). On CUB-200-2011 dataset for GoogLeNet architecture, with Maximum-Entropy, the mean probability of the top class is 0.34, as compared to 0.77 without it. Moreover, the tail of probability values is fatter with Maximum-Entropy, as depicted in Figure 2b.

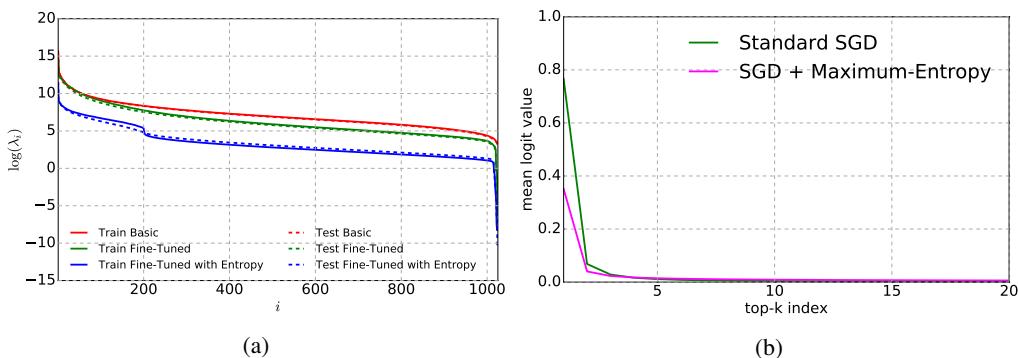

(a)             (b)

Figure 2: (a) Maximum-Entropy training encourages the network to reduce the specificity of the features, which is reflected in the longer tail of eigenvalues for the covariance matrix of `pool5` GoogLeNet features for both training and test sets of CUB-200-2011. We plot the value of $\log(\lambda_i)$ for the $i$th eigenvalue $\lambda_i$ obtained after decomposition of test set (dashed) and training set (solid) (for $\gamma = 1$). (b) For Maximum-Entropy training, the predicted logit vector is smoother with a fatter tail (GoogLeNet on CUB-200-2011).

## 4.2 Ablation Studies

**CIFAR-10 and CIFAR-100:** We evaluate Maximum-Entropy on the CIFAR-10 and CIFAR-100 datasets [23]. CIFAR-100 has the same set of images as CIFAR-10 but with finer category distinction in the labels, with each "superclass" of 20 containing five finer divisions, and a 100 categories in total. Therefore, we expect (and observe) that Maximum-Entropy training provides stronger gains on CIFAR-100 as compared to CIFAR-10 across models (Table 2).

| Method | | CUB-200-2011 | Cars | Aircrafts | NABirds | Stanford Dogs |
|---|---|---|---|---|---|---|
| VGG-Net16 | **MaxEnt** | 77.02 | 83.88 | 78.08 | 72.62 | 65.44 |
| | LSR | 70.03 | 81.45 | 75.06 | 69.28 | 63.06 |
| ResNet-50 | **MaxEnt** | 80.37 | 93.85 | 83.86 | 69.21 | 73.56 |
| | LSR | 78.20 | 92.04 | 81.26 | 64.02 | 70.03 |
| DenseNet-161 | **MaxEnt** | 86.54 | 93.01 | 89.76 | 83.02 | 83.63 |
| | LSR | 84.86 | 91.96 | 87.05 | 80.11 | 82.98 |

Table 4: Maximum-Entropy training obtains much large gains on Fine-grained Visual Classification as compared to Label Smoothing Regularization (LSR) [40].

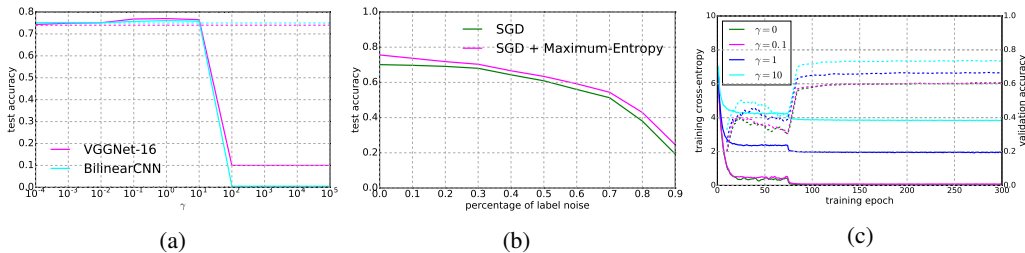

(a)  (b)  (c)

Figure 3: (a) Classification performance is robust to the choice of $\gamma$ over a large region as shown here for CUB-200-2011 with models VGGNet-16 and BilinearCNN. (b) Maximum-Entropy is more robust to increasing amounts of label noise (CUB-200-2011 on GoogleNet with $\gamma = 1$). (c) Maximum-Entropy obtains higher validation performance despite higher training cross-entropy loss.

**ImageNet Ablation Experiment:** To understand the effect of Maximum-Entropy training on datasets with more samples compared to the small fine-grained datasets, we create two synthetic datasets: (i) Random-ImageNet, which is formed by selecting 116K images from a random subset of 117 classes of ImageNet [7], and (ii) Dogs-ImageNet, which is formed by selecting all classes from ImageNet that have dogs as labels, which has the same number of images and classes as Random-ImageNet. Dogs-ImageNet has less diversity compared to Random-ImageNet, and thus we expect the gains from Maximum-Entropy to be higher. On a 5-way cross-validation on both dataset, we observe higher gains on the Dogs-ImageNet dataset for two CNN models (Table 3).

**Choice of Hyperparameter $\gamma$:** An integral component of regularization is the choice of weighing parameter. We find that performance is fairly robust to the choice of $\gamma$ (Figure 3a). Please see supplement for experiment-wise details.

**Robustness to Label Noise:** In this experiment, we gradually introduce label noise by randomly permuting a fraction of labels for increasing fractions of total data. We follow an identical evaluation protocol as the previous experiment, and observe that Maximum-Entropy is more robust to label noise (Figure 3b).

**Training Cross-Entropy and Validation Accuracy:** We expect Maximum-Entropy training to provide higher accuracy at the cost of higher training cross-entropy. In Figure 3c, we show that we achieve a higher validation accuracy when training with Maximum-Entropy despite the training cross-entropy loss converging to a higher value.

**Comparison with Label-Smoothing Regularization:** Label-Smoothing Regularization [40] penalizes the KL-divergence of the classifier logits from the uniform distribution – and is also a method to prevent peaky distributions. On comparing performance with Label-Smoothing Regularization, we found that Maximum-Entropy provides much larger gains on fine-grained recognition (see Table 4).

## 5   Discussion and Conclusion

Many real-world applications of computer vision models involve extensive fine-tuning on small, relatively imbalanced datasets with much smaller diversity in the training set compared to the large-scale models they are fine-tuned from, a notable example of which is fine-grained recognition. In this domain, Maximum-Entropy training provides an easy-to-implement and simple to understand training schedule that consistently improves performance. There are several extensions, however, that

can be explored: explicitly enforcing a large diversity in the features through a different regularizer might be an interesting extension to this study, as well as potential extensions to large-scale problems by tackling clusters of diverse objects separately. We leave these as a future study with our results as a starting point.

**Acknowledgements:** We thank Ryan Farrell, Pei Guo, Xavier Boix, Dhaval Adjodah, Spandan Madan, and Ishaan Grover for their feedback on the project and Google's TensorFlow Research Cloud Program for providing TPU computing resources.

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
