[Supplementary Material · supplement.pdf]

# Appendix 1: Preliminaries

## Probabilistic Tail Bounds

**Theorem 1** (Hoeffding's Inequality (Theorem 2.8 of [1])). *Let $X_1, \ldots, X_n$ be independent random variables such that $X_i$ takes its values in $[a_i, b_i]$ almost surely for all $i \leq n$. Let*

$$S = \sum_{i=1}^{n} \left( X_i - \mathbb{E}\left[ X_i \right] \right),$$

*then for every $t > 0$,*

$$\Pr\left( S \geq t \right) \leq \exp\left( -\frac{2n^2 t^2}{\sum_{i=1}^{n}(a_i - b_i)^2} \right).$$

**Theorem 2** (Cantelli's Inequality (Equation 7 of [2])). *The inequality states that*

$$\Pr(X - \mu \geq \lambda) \quad \begin{cases} \leq \frac{\sigma^2}{\sigma^2 + \lambda^2} & \text{if } \lambda > 0, \\ \geq 1 - \frac{\sigma^2}{\sigma^2 + \lambda^2} & \text{if } \lambda < 0. \end{cases}$$

*where $X$ is a real-valued random variable, $\Pr$ is the probability measure, $\mu$ is the expected value of $X$, $\sigma^2$ is the variance of $X$.*

## Basic Derivations for Multivariate Gaussian Mixtures

**Lemma 1.** *For $N$ vectors $\mathbf{x}_1, \ldots, \mathbf{x}_N$, $\mathbf{x}_i \in \mathbb{R}^m \; \forall i$, $N$ constants $\alpha_1, \ldots, \alpha_N$, $\alpha_i > 0 \; \forall i$, $\sum_{i=1}^{N} \alpha_i = 1$ and target vector $\mathbf{y} \in \mathbb{R}^m$,*

$$\sum_{i=1}^{N} \alpha_i \mathbf{x}_i^\top \mathbf{y} \leq \left( \max_i \|\mathbf{x_i}\|_2 \right) \cdot \|\mathbf{y}\|_2$$

*Proof.* For each vector $\mathbf{x}_i$, we know by the Cauchy-Schwarz Inequality that:

$$\mathbf{x}_i^\top \mathbf{y} \leq \|\mathbf{x}_i\|_2 \cdot \|\mathbf{y}\|_2 \tag{1}$$

And:

$$\|\mathbf{x}_k\|_2 \leq \max_i \|\mathbf{x_i}\|_2 \; \forall k \tag{2}$$

Combining the above, we have:

$$\sum_{i=1}^{N} \alpha_i \mathbf{x}_i^\top \mathbf{y} \leq \sum_{i=1}^{N} \alpha_i \|\mathbf{x}_i\|_2 \cdot \|\mathbf{y}\|_2 \leq (\sum_{i=1}^{N} \alpha_i) \left( \max_i \|\mathbf{x_i}\|_2 \right) \cdot \|\mathbf{y}\|_2 = \left( \max_i \|\mathbf{x_i}\|_2 \right) \cdot \|\mathbf{y}\|_2 \tag{3}$$

$\square$

**Lemma 2.** *For $N$ vectors $\mathbf{x}_1, \ldots, \mathbf{x}_N$, $\mathbf{x}_i \in \mathbb{R}^m \; \forall i$, and target vector $\mathbf{y} \in \mathbb{R}^m$,*

$$\sum_{i=1}^{N} \mathbf{x}_i^\top \mathbf{y} \geq -N \left( \max_i \|\mathbf{x_i}\|_2 \right) \cdot \|\mathbf{y}\|_2$$

*Proof.* For each vector $\mathbf{x}_i$, we know by the Cauchy-Schwarz Inequality that:

$$-\mathbf{x}_i^\top \mathbf{y} \leq \|-\mathbf{x}_i\|_2 \cdot \|\mathbf{y}\|_2 \tag{4}$$
$$= \|\mathbf{x}_i\|_2 \cdot \|\mathbf{y}\|_2 \tag{5}$$

Multiplying the above equation by $-1$, we have:

$$\mathbf{x}_i^\top \mathbf{y} \geq -\|\mathbf{x}_i\|_2 \cdot \|\mathbf{y}\|_2 \tag{6}$$

And:

$$\|\mathbf{x}_k\|_2 \leq \max_i \|\mathbf{x_i}\|_2 \; \forall k \tag{7}$$

Multiplying the above equation by $-1$, we have:

$$-\|\mathbf{x}_k\|_2 \geq -\max_i \|\mathbf{x_i}\|_2 \; \forall k \tag{8}$$

Combining the above, we have:

$$\sum_{i=1}^{N} \mathbf{x}_i^\top \mathbf{y} \geq -\sum_{i=1}^{N} \|\mathbf{x}_i\|_2 \cdot \|\mathbf{y}\|_2 \geq -N\left(\max_i \|\mathbf{x_i}\|_2\right) \cdot \|\mathbf{y}\|_2 \tag{9}$$

$\square$

**Lemma 3.** *For an $n$-dimensional multivariate normal distribution $X \sim \mathcal{N}(\mu, \Sigma)$, we have:*

$$\mathbb{E}[\|X\|_2^2] = \text{tr}(\Sigma) + \|\mu\|_2^2$$

*Proof.*

$$\mathbb{E}[\|X\|_2^2] = \sum_{i=1}^{n} \mathbb{E}[X_i^2] = \sum_{i=1}^{n} \left(\text{Var}[X_i] + (\mathbb{E}[X_i])^2\right) = \text{tr}(\Sigma) + \sum_{i=1}^{n} \mathbb{E}[X_i]^2 = \text{tr}(\Sigma) + \|\mu\|_2^2 \tag{10}$$

$\square$

**Lemma 4.** *For a random variable $X$ that is distributed by an $n$-dimensional mixture of $m$ Gaussians, that is $X \sim \sum_{i=1}^{m} \alpha_i \mathcal{N}(\mu_i, \Sigma_i)$ for $\alpha_i > 0 \; \forall i$ and $\sum_{i=1}^{m} \alpha_i = 1$:*

$$\mathbb{E}[\|X\|_2^2] = \sum_{i=1}^{m} \alpha_i(\text{tr}(\Sigma_i) + \|\mu_i\|_2^2)$$

*Proof.* By law of conditional expectation:

$$\mathbb{E}[\|X\|_2^2] = \sum_{i=1}^{m} \mathbb{E}[\mathbb{E}[\|X\|_2^2|i]] = \sum_{i=1}^{m} \alpha_i \mathbb{E}[\|X\|_2^2|i] \tag{11}$$

Since the conditional distribution given the mixture component $i$ is $n$-dimensional Gaussian $\mathcal{N}(\mu_i, \Sigma_i)$, from Lemma 3, we have:

$$= \sum_{i=1}^{m} \alpha_i(\text{tr}(\Sigma_i) + \|\mu_i\|_2^2) \tag{12}$$

$\square$

**Classification Preliminaries**

Consider the multi-class classification problem over $m$ classes. The input domain is given by $\mathcal{X} \subset \mathbb{R}^Z$, with an accompanying probability metric $p_\mathsf{x}(\cdot)$ defined over $\mathcal{X}$. The training data is given by $N$ i.i.d. samples $\mathcal{D} = \{\mathbf{x}_1, ..., \mathbf{x}_N\}$ drawn from $\mathcal{X}$. Each point $\mathbf{x} \in \mathcal{X}$ has an associated label $\bar{\mathbf{y}}(\mathbf{x}) = [0, ..., 1, ...0] \in \mathbb{R}^m$. We learn a CNN such that for each point in $\mathcal{X}$, the CNN induces a conditional probability distribution over the $m$ classes whose mode matches the label $\bar{\mathbf{y}}(\mathbf{x})$.

A CNN architecture consists of a series of convolutional and subsampling layers that culminate in an *activation* $\Phi(\cdot)$, which is fed to an $m$-way classifier with weights $\mathbf{w} = \{\mathbf{w}_1, ..., \mathbf{w}_m\}$ such that:

$$p(y_i|\mathbf{x}; \mathbf{w}, \Phi(\cdot)) = \frac{\exp\left(\mathbf{w}_i^\top \Phi(\mathbf{x})\right)}{\sum_{j=1}^{m} \exp\left(\mathbf{w}_j^\top \Phi(\mathbf{x})\right)} \tag{13}$$

The entropy of conditional probability distribution in Equation 13 is given by:

$$\mathsf{H}[p(\cdot|\mathbf{x};\boldsymbol{\theta})] \triangleq -\sum_{i=1}^{m} p(y_i|\mathbf{x};\boldsymbol{\theta}) \log(p(y_i|\mathbf{x};\boldsymbol{\theta})) \tag{14}$$

The expected entropy over the distribution is given by:

$$\mathbb{E}_{\mathbf{x}\sim p_{\mathsf{x}}}\left[\mathsf{H}[p(\cdot|\mathbf{x};\boldsymbol{\theta})]\right] = \int_{\mathbf{x}\sim p_{\mathsf{x}}} \mathsf{H}[p(\cdot|\mathbf{x};\boldsymbol{\theta})]p_{\mathsf{x}}(\mathbf{x})d\mathbf{x} \tag{15}$$

The empirical average of the conditional entropy over the training set $\mathcal{D}$ is:

$$\hat{\mathbb{E}}_{\mathbf{x}\sim\mathcal{D}}[\mathsf{H}[p(\cdot|\mathbf{x};\boldsymbol{\theta})]] = \frac{1}{N}\sum_{i=1}^{N} \mathsf{H}[p(\cdot|\mathbf{x}_i;\boldsymbol{\theta})] \tag{16}$$

*Diversity $\boldsymbol{\nu}(\Phi, p_{\mathsf{x}})$ of the features is given by:*

$$\boldsymbol{\nu}(\Phi, p_{\mathsf{x}}) \triangleq \sum_{i=1}^{n} \lambda_i = \mathsf{tr}(\boldsymbol{\Sigma}^*) = \sum_{i=1}^{m} \alpha_i \left(\mathsf{tr}(\boldsymbol{\Sigma}_i) + \mathsf{tr}(\boldsymbol{\mu}_i\boldsymbol{\mu}_i^{\top})\right) = \sum_{i=1}^{m} \alpha_i \left(\mathsf{tr}(\boldsymbol{\Sigma}_i) + \|\boldsymbol{\mu}_i\|_2^2\right) \tag{17}$$

## Appendix 2: Theoretical Results

**Lemma 5.** *For the above classification setup, where $\|\mathbf{w}\|_{\infty} = \max_i (\|\mathbf{w}_i\|_2)$ :*

$$\mathsf{H}[p(\cdot|\mathbf{x};\mathbf{w})] \geq \log(C) - 2\|\mathbf{w}\|_{\infty}\|\Phi(\mathbf{x})\|_2 \tag{18}$$

*Proof.* For an input $\mathbf{x}$, the conditional probability distribution over $m$ classes for a statistical model with feature map $\Phi(\mathbf{x})$ and weights $\mathbf{w} = (\mathbf{w}_1, ..., \mathbf{w}_C)$ can be given by:

$$p(y_i|\mathbf{x};\mathbf{w}) = \frac{\exp\left(\mathbf{w}_i^{\top}\Phi(\mathbf{x})\right)}{\sum_{j=1}^{C}\exp\left(\mathbf{w}_j^{\top}\Phi(\mathbf{x})\right)} \tag{19}$$

We can thus write the conditional entropy $\mathsf{H}[p(\cdot|\mathbf{x};\mathbf{w})]$ for the above sample as:

$$\mathsf{H}[p(\cdot|\mathbf{x};\mathbf{w})] = -\sum_{i=1}^{C} p(y_i|\mathbf{x};\mathbf{w}) \log\left(p(y_i|\mathbf{x};\mathbf{w})\right) \tag{20}$$

$$= -\sum_{i=1}^{C}\left(\frac{\exp\left(\mathbf{w}_i^{\top}\Phi(\mathbf{x})\right)}{\sum_{j=1}^{C}\exp\left(\mathbf{w}_j^{\top}\Phi(\mathbf{x})\right)} \cdot \left(\mathbf{w}_i^{\top}\Phi(\mathbf{x}) - \log\left(\sum_{j=1}^{C}\exp\left(\mathbf{w}_j^{\top}\Phi(\mathbf{x})\right)\right)\right)\right) \tag{21}$$

$$= \log\left(\sum_{j=1}^{C}\exp\left(\mathbf{w}_j^{\top}\Phi(\mathbf{x})\right)\right) - \frac{\sum_{i=1}^{C}\left(\exp\left(\mathbf{w}_i^{\top}\Phi(\mathbf{x})\right) \cdot \mathbf{w}_i^{\top}\Phi(\mathbf{x})\right)}{\sum_{j=1}^{C}\exp\left(\mathbf{w}_j^{\top}\Phi(\mathbf{x})\right)} \tag{22}$$

$$= \log(m) + \log\left(\frac{1}{C}\sum_{j=1}^{C}\exp\left(\mathbf{w}_j^{\top}\Phi(\mathbf{x})\right)\right) - \frac{\sum_{i=1}^{C}\left(\exp\left(\mathbf{w}_i^{\top}\Phi(\mathbf{x})\right) \cdot \mathbf{w}_i^{\top}\Phi(\mathbf{x})\right)}{\sum_{j=1}^{C}\exp\left(\mathbf{w}_j^{\top}\Phi(\mathbf{x})\right)} \tag{23}$$

Since $\log$ is a concave function:

$$\geq \log(C) + \frac{1}{C}\sum_{j=1}^{m}\left(\mathbf{w}_j^{\top}\Phi(\mathbf{x})\right) - \frac{\sum_{i=1}^{C}\left(\exp\left(\mathbf{w}_i^{\top}\Phi(\mathbf{x})\right) \cdot \mathbf{w}_i^{\top}\Phi(\mathbf{x})\right)}{\sum_{j=1}^{C}\exp\left(\mathbf{w}_j^{\top}\Phi(\mathbf{x})\right)} \tag{24}$$

By Lemma 1, we have:

$$\geq \log(C) + \frac{1}{C}\sum_{j=1}^{C}\left(\mathbf{w}_j^{\top}\Phi(\mathbf{x})\right) - \|\mathbf{w}\|_{\infty}\|\Phi(\mathbf{x})\|_2 \tag{25}$$

By Lemma 2, we have:

$$\geq \log(C) - 2\|\mathbf{w}\|_{\infty}\|\Phi(\mathbf{x})\|_2 \tag{26}$$

$\square$

Now we are ready to prove Theorem 1 from the main paper.

**Theorem 3** (Theorem 1 from Text: Lower Bound on $\ell_2$-norm of Classifier). *The expected conditional entropy follows:*

$$\|\mathbf{w}\|_2 \geq \frac{\log(C) - \mathbb{E}_{\mathbf{x}\sim p_{\mathsf{x}}}[\mathsf{H}[p(\cdot|\mathbf{x};\boldsymbol{\theta})]]}{2\sqrt{\boldsymbol{\nu}(\Phi, p_{\mathsf{x}})}}$$

*Proof.* From Lemma 5, we have:

$$\mathsf{H}[p(\cdot|\mathbf{x};\mathbf{w})] \geq \log(C) - 2\|\mathbf{w}\|_{\infty}\|\Phi(\mathbf{x})\|_2 \tag{27}$$

Since $\|\mathbf{w}\|_2 = \sqrt{\sum_{i=1}^{C}\|\mathbf{w}_i\|_2^2} \geq \|\mathbf{w}\|_{\infty}$, we have:

$$\mathsf{H}[p(\cdot|\mathbf{x};\mathbf{w})] \geq \log(C) - 2\|\mathbf{w}\|_2\|\Phi(\mathbf{x})\|_2 \tag{28}$$

Taking expectation over $p_{\mathsf{x}}$, we have:

$$\mathbb{E}_{\mathbf{x}\sim p_{\mathsf{x}}}[\mathsf{H}[p(\cdot|\mathbf{x};\mathbf{w})]] \geq \log(C) - 2\|\mathbf{w}\|_2\mathbb{E}_{\mathbf{x}\sim p_{\mathsf{x}}}[\|\Phi(\mathbf{x})\|_2] \tag{29}$$

By Cauchy-Schwarz Inequality, $\mathbb{E}_{\mathbf{x}\sim p_{\mathsf{x}}}[\|\Phi(\mathbf{x})\|_2] \leq \sqrt{\mathbb{E}_{\mathbf{x}\sim p_{\mathsf{x}}}[\|\Phi(\mathbf{x})\|_2^2]}$. Using this:

$$\geq \log(C) - 2\|\mathbf{w}\|_2\sqrt{\mathbb{E}_{\mathbf{x}\sim p_{\mathsf{x}}}[\|\Phi(\mathbf{x})\|_2^2]} \tag{30}$$

By Lemma 4, we have:

$$= \log(C) - 2\|\mathbf{w}\|_2\sqrt{\sum_{i=1}^{m}\alpha_i(\mathsf{tr}(\boldsymbol{\Sigma}_i) + \|\mu_i\|_2^2)} \tag{31}$$

Rearranging and using the definition of *Diversity* we have:

$$\|\mathbf{w}\|_2 \geq \frac{\log(C) - \mathbb{E}_{\mathbf{x}\sim p_{\mathsf{x}}}[\mathsf{H}[p(\cdot|\mathbf{x};\boldsymbol{\theta})]]}{2\sqrt{\boldsymbol{\nu}(\Phi, p_{\mathsf{x}})}} \tag{32}$$

$\square$

**Lemma 6.** *With probability at least $1 - \delta/2$,*

$$\left|\hat{\mathbb{E}}_{\mathcal{D}}[\mathsf{H}[p(\cdot|\mathbf{x};\boldsymbol{\theta})]] - \mathbb{E}_{\mathbf{x}\sim p_{\mathsf{x}}}[\mathsf{H}[p(\cdot|\mathbf{x};\boldsymbol{\theta})]]\right| \leq \|\mathbf{w}\|_{\infty}\sqrt{\frac{2\hat{\mathbb{E}}_{\mathbf{x}\sim\mathcal{D}}[\|\Phi(\mathbf{x})\|_2^2]}{N}\log(\frac{4}{\delta})}$$

*Proof.* Since $\mathcal{D}$ has i.i.d. samples of $\mathcal{X}$, we have:

$$\mathbb{E}_{\mathbf{x}\sim p_{\mathsf{x}}}\left[\hat{\mathbb{E}}_{\mathcal{D}}\left[\mathsf{H}\left[p(\cdot|\mathbf{x};\mathbf{w})\right]\right]\right] = \frac{1}{N}\sum_{i=1}^{N}\mathbb{E}_{\mathbf{x}\sim p_{\mathsf{x}}}\left[\mathsf{H}\left[p(\cdot|\mathbf{x}_i;\mathbf{w})\right]\right] = \mathbb{E}_{\mathbf{x}\sim p_{\mathsf{x}}}\left[\mathsf{H}\left[p(\cdot|\mathbf{x};\mathbf{w})\right]\right] \tag{33}$$

From Lemma 5, we know that for sample $\mathbf{x}$:

$$\log(m) - 2\|\mathbf{w}\|_{\infty}\|\Phi(\mathbf{x})\|_2 \leq \mathsf{H}\left[p(\cdot|\mathbf{x};\mathbf{w})\right] \leq \log(m) \tag{34}$$

Thus, by applying Hoeffding's Inequality we get:

$$\Pr\left(\left|\hat{\mathbb{E}}_{\mathcal{D}}[\mathsf{H}[p(\cdot|\mathbf{x};\boldsymbol{\theta})]] - \mathbb{E}_{\mathbf{x}\sim p_{\mathsf{x}}}\left[\hat{\mathbb{E}}_{\mathcal{D}}\left[\mathsf{H}\left[p(\cdot|\mathbf{x};\mathbf{w})\right]\right]\right]\right| \geq t\right) \leq 2\exp\frac{-2N^2t^2}{4\|\mathbf{w}\|_\infty^2 \sum_{i=1}^N \|\Phi(\mathbf{x}_i)\|^2} \tag{35}$$

Setting RHS as $\delta/2$, we have with probability at least $1 - \delta/2$:

$$\left|\hat{\mathbb{E}}_{\mathcal{D}}[\mathsf{H}[p(\cdot|\mathbf{x};\boldsymbol{\theta})]] - \mathbb{E}_{\mathbf{x}\sim p_{\mathsf{x}}}[\mathsf{H}[p(\cdot|\mathbf{x};\boldsymbol{\theta})]]\right| \leq \|\mathbf{w}\|_\infty \sqrt{\frac{2\hat{\mathbb{E}}_{\mathbf{x}\sim\mathcal{D}}[\|\Phi(\mathbf{x})\|_2^2]}{N}\log(\frac{4}{\delta})} \tag{36}$$

$\square$

**Lemma 7.** *With probability at least* $1 - \delta/2$, *we have:*

$$\hat{\mathbb{E}}_{\mathbf{x}\sim\mathcal{D}}[\|\Phi(\mathbf{x})\|_2^2] \leq \boldsymbol{\nu}(\Phi, p_{\mathsf{x}}) + \sqrt{\frac{\mathsf{Var}_{p_{\mathsf{x}}}[\|\Phi(\mathbf{x})\|_2^2](2/\delta - 1)}{N}} \tag{37}$$

*Proof.* Since $\mathcal{D}$ has i.i.d. samples of $\mathcal{X}$,:

$$\mathsf{Var}_{p_{\mathsf{x}}}[\hat{\mathbb{E}}_{\mathbf{x}\sim\mathcal{D}}[\|\Phi(\mathbf{x})\|_2^2]] = \frac{1}{N^2}\sum_{i=1}^m \mathsf{Var}_{p_{\mathsf{x}}}[\|\Phi(\mathbf{x}_i)\|_2^2] = \frac{\mathsf{Var}_{p_{\mathsf{x}}}[\|\Phi(\mathbf{x})\|_2^2]}{N} \tag{38}$$

Now, by the Cantelli Inequality, we have for $t > 0$:

$$\Pr\left(\hat{\mathbb{E}}_{\mathbf{x}\sim\mathcal{D}}[\|\Phi(\mathbf{x})\|_2^2] < \mathbb{E}_{\mathbf{x}\sim p_{\mathsf{x}}}[\|\Phi(\mathbf{x})\|_2^2] + t\right) \geq 1 - \left(1 + \frac{t^2}{\mathsf{Var}_{p_{\mathsf{x}}}[\hat{\mathbb{E}}_{\mathbf{x}\sim\mathcal{D}}[\|\Phi(\mathbf{x})\|_2^2]]}\right)^{-1} \tag{39}$$

Setting RHS ast $1 - \delta/2$, we have and solving for $t$, we have with probability at least $1 - \delta/2$:

$$\hat{\mathbb{E}}_{\mathbf{x}\sim\mathcal{D}}[\|\Phi(\mathbf{x})\|_2^2] \leq \hat{\mathbb{E}}_{\mathbf{x}\sim p_{\mathsf{x}}}[\|\Phi(\mathbf{x})\|_2^2] + \sqrt{\frac{\mathsf{Var}_{p_{\mathsf{x}}}[\|\Phi(\mathbf{x})\|_2^2](2/\delta - 1)}{N}} \tag{40}$$

Using the result from Lemma 4 and the definition of *Diversity*, we have with probability at least $1 - \delta/2$:

$$\hat{\mathbb{E}}_{\mathbf{x}\sim\mathcal{D}}[\|\Phi(\mathbf{x})\|_2^2] \leq \boldsymbol{\nu}(\Phi, p_{\mathsf{x}}) + \sqrt{\frac{\mathsf{Var}_{p_{\mathsf{x}}}[\|\Phi(\mathbf{x})\|_2^2](2/\delta - 1)}{N}} \tag{41}$$

$\square$

**Theorem 4** (Theorem 2 from Main Text: Uniform Convergence of Entropy Estimate). *With probability at least* $1 - \delta$,

$$\left|\hat{\mathbb{E}}_{\mathcal{D}}[\mathsf{H}[p(\cdot|\mathbf{x};\boldsymbol{\theta})]] - \mathbb{E}_{\mathbf{x}\sim p_{\mathsf{x}}}[\mathsf{H}[p(\cdot|\mathbf{x};\boldsymbol{\theta})]]\right| \leq \|\mathbf{w}\|_\infty\left(\sqrt{\frac{2}{N}\boldsymbol{\nu}(\Phi, p_{\mathsf{x}})\log(\frac{4}{\delta})} + \boldsymbol{\Theta}\left(N^{-0.75}\right)\right)$$

*Proof.* From Lemma 6, we have with probability at least $1 - \delta/2$:

$$\left|\hat{\mathbb{E}}_{\mathcal{D}}[\mathsf{H}[p(\cdot|\mathbf{x};\boldsymbol{\theta})]] - \mathbb{E}_{\mathbf{x}\sim p_{\mathsf{x}}}[\mathsf{H}[p(\cdot|\mathbf{x};\boldsymbol{\theta})]]\right| \leq \|\mathbf{w}\|_\infty \sqrt{\frac{2\hat{\mathbb{E}}_{\mathbf{x}\sim\mathcal{D}}[\|\Phi(\mathbf{x})\|_2^2]}{N}\log(\frac{4}{\delta})} \tag{42}$$

From Lemma 7, we also have with probability at least $1 - \delta/2$:

$$\hat{\mathbb{E}}_{\mathbf{x}\sim\mathcal{D}}[\|\Phi(\mathbf{x})\|_2^2] \leq \boldsymbol{\nu}(\Phi, p_{\mathsf{x}}) + \sqrt{\frac{\mathsf{Var}_{p_{\mathsf{x}}}[\|\Phi(\mathbf{x})\|_2^2](2/\delta - 1)}{N}} \tag{43}$$

Combining the above two statements using the Union Bound, we have with probability at least $1 - \delta$:

$$\left|\hat{\mathbb{E}}_{\mathcal{D}}[\mathsf{H}[p(\cdot|\mathbf{x};\boldsymbol{\theta})]] - \mathbb{E}_{\mathbf{x}\sim p_{\mathsf{x}}}[\mathsf{H}[p(\cdot|\mathbf{x};\boldsymbol{\theta})]]\right| \leq \|\mathbf{w}\|_{\infty}\sqrt{\frac{2}{N}(\boldsymbol{\nu}(\Phi,p_{\mathsf{x}}) + \sqrt{\frac{\mathsf{Var}_{p_{\mathsf{x}}}[\|\Phi(\mathbf{x})\|_2^2](2/\delta - 1)}{N}})\log(\frac{4}{\delta})}$$
(44)

$$\left|\hat{\mathbb{E}}_{\mathcal{D}}[\mathsf{H}[p(\cdot|\mathbf{x};\boldsymbol{\theta})]] - \mathbb{E}_{\mathbf{x}\sim p_{\mathsf{x}}}[\mathsf{H}[p(\cdot|\mathbf{x};\boldsymbol{\theta})]]\right| \leq \|\mathbf{w}\|_{\infty}\left(\sqrt{\frac{2}{N}(\boldsymbol{\nu}(\Phi,p_{\mathsf{x}})\log(\frac{4}{\delta})} + \left(\frac{4\mathsf{Var}_{p_{\mathsf{x}}}[\|\Phi(\mathbf{x})\|_2^2](2/\delta - 1)}{N^3}\right)^{1/4}\log(\frac{4}{\delta})\right)$$
(45)

$$\left|\hat{\mathbb{E}}_{\mathcal{D}}[\mathsf{H}[p(\cdot|\mathbf{x};\boldsymbol{\theta})]] - \mathbb{E}_{\mathbf{x}\sim p_{\mathsf{x}}}[\mathsf{H}[p(\cdot|\mathbf{x};\boldsymbol{\theta})]]\right| \leq \|\mathbf{w}\|_{\infty}\left(\sqrt{\frac{2}{N}\boldsymbol{\nu}(\Phi,p_{\mathsf{x}})\log(\frac{4}{\delta})} + \boldsymbol{\Theta}\left(N^{-0.75}\right)\right)$$
(46)

$\square$

**Lemma 8.** *With probability at least $1 - \delta/2$,*

$$\hat{\mathbb{E}}_{\mathcal{D}}[\mathsf{H}[p(\cdot|\mathbf{x};\boldsymbol{\theta})]] \leq \mathbb{E}_{\mathbf{x}\sim p_{\mathsf{x}}}[\mathsf{H}[p(\cdot|\mathbf{x};\boldsymbol{\theta})]] + \|\mathbf{w}\|_2\sqrt{\frac{2\hat{\mathbb{E}}_{\mathbf{x}\sim\mathcal{D}}[\|\Phi(\mathbf{x})\|_2^2]}{N}\log(\frac{2}{\delta})}$$

*Proof.* Since $\mathcal{D}$ has i.i.d. samples of $\mathcal{X}$, we have:

$$\mathbb{E}_{\mathbf{x}\sim p_{\mathsf{x}}}\left[\hat{\mathbb{E}}_{\mathcal{D}}\left[\mathsf{H}\left[p(\cdot|\mathbf{x};\mathbf{w})\right]\right]\right] = \frac{1}{N}\sum_{i=1}^{N}\mathbb{E}_{\mathbf{x}\sim p_{\mathsf{x}}}\left[\mathsf{H}\left[p(\cdot|\mathbf{x}_i;\mathbf{w})\right]\right] = \mathbb{E}_{\mathbf{x}\sim p_{\mathsf{x}}}\left[\mathsf{H}\left[p(\cdot|\mathbf{x};\mathbf{w})\right]\right]$$
(47)

From Lemma 5, we know that for sample $\mathbf{x}$:

$$\log(m) - 2\|\mathbf{w}\|_2\|\Phi(\mathbf{x})\|_2 \leq \mathsf{H}\left[p(\cdot|\mathbf{x};\mathbf{w})\right] \leq \log(m)$$
(48)

Thus, by applying one-sided Hoeffding's Inequality we get:

$$\mathrm{Pr}\left(\hat{\mathbb{E}}_{\mathcal{D}}[\mathsf{H}[p(\cdot|\mathbf{x};\boldsymbol{\theta})]] - \mathbb{E}_{\mathbf{x}\sim p_{\mathsf{x}}}\left[\hat{\mathbb{E}}_{\mathcal{D}}\left[\mathsf{H}\left[p(\cdot|\mathbf{x};\mathbf{w})\right]\right]\right] \geq t\right) \leq \exp\frac{-2N^2t^2}{4\|\mathbf{w}\|_2^2\sum_{i=1}^{N}\|\Phi(\mathbf{x}_i)\|^2}$$
(49)

Setting RHS as $\delta/2$, we have with probability at least $1 - \delta/2$:

$$\hat{\mathbb{E}}_{\mathcal{D}}[\mathsf{H}[p(\cdot|\mathbf{x};\boldsymbol{\theta})]] \leq \mathbb{E}_{\mathbf{x}\sim p_{\mathsf{x}}}[\mathsf{H}[p(\cdot|\mathbf{x};\boldsymbol{\theta})]] + \|\mathbf{w}\|_2\sqrt{\frac{2\hat{\mathbb{E}}_{\mathbf{x}\sim\mathcal{D}}[\|\Phi(\mathbf{x})\|_2^2]}{N}\log(\frac{2}{\delta})}$$
(50)

$\square$

**Corollary 1** (Corollary 1 from the Main Text: Theorem 1 in terms of Variance of Norm). *With probability at least $1 - \delta$,*

$$\|\mathbf{w}\|_2 \geq \frac{\log(C) - \hat{\mathbb{E}}_{\mathbf{x}\sim\mathcal{D}}[\mathsf{H}[p(\cdot|\mathbf{x};\boldsymbol{\theta})]]}{\left(2 - \sqrt{\frac{2}{N}\log(\frac{2}{\delta})}\right)\sqrt{\boldsymbol{\nu}(\Phi,p_{\mathsf{x}})} - \boldsymbol{\Theta}\left(N^{-0.75}\right)}$$

*Proof.* From Lemma 8, we have with probability at least $1 - \delta/2$:

$$\hat{\mathbb{E}}_{\mathcal{D}}[\mathsf{H}[p(\cdot|\mathbf{x};\boldsymbol{\theta})]] \leq \mathbb{E}_{\mathbf{x}\sim p_{\mathsf{x}}}[\mathsf{H}[p(\cdot|\mathbf{x};\boldsymbol{\theta})]] + \|\mathbf{w}\|_2\sqrt{\frac{2\hat{\mathbb{E}}_{\mathbf{x}\sim\mathcal{D}}[\|\Phi(\mathbf{x})\|_2^2]}{N}\log(\frac{2}{\delta})}$$
(51)

From Lemma 7, we also have with probability at least $1 - \delta/2$:

$$\hat{\mathbb{E}}_{\mathbf{x}\sim\mathcal{D}}[\|\Phi(\mathbf{x})\|_2^2] \leq \boldsymbol{\nu}(\Phi,p_{\mathsf{x}}) + \sqrt{\frac{\mathsf{Var}_{p_{\mathsf{x}}}[\|\Phi(\mathbf{x})\|_2^2](2/\delta - 1)}{N}}$$
(52)

Combining the above two statements using the Union Bound, we have with probability at least $1 - \delta$:

$$\hat{\mathbb{E}}_{\mathcal{D}}[\mathsf{H}[p(\cdot|\mathbf{x};\boldsymbol{\theta})]] \leq \mathbb{E}_{\mathbf{x}\sim p_{\mathsf{x}}}[\mathsf{H}[p(\cdot|\mathbf{x};\boldsymbol{\theta})]] + \|\mathbf{w}\|_2 \sqrt{\frac{2}{N}\left(\boldsymbol{\nu}(\Phi, p_{\mathsf{x}}) + \sqrt{\frac{\mathsf{Var}_{p_{\mathsf{x}}}[\|\Phi(\mathbf{x})\|_2^2](2/\delta - 1)}{N}}\right)\log(\frac{2}{\delta})} \tag{53}$$

From Theorem **??**, we know:

$$\|\mathbf{w}\|_2 \geq \frac{\log(C) - \mathbb{E}_{\mathbf{x}\sim p_{\mathsf{x}}}[\mathsf{H}[p(\cdot|\mathbf{x};\boldsymbol{\theta})]]}{2\sqrt{\boldsymbol{\nu}(\Phi, p_{\mathsf{x}})}} \tag{54}$$

Combining this with the previous statement, we have with probability at least $1 - \delta$:

$$\|\mathbf{w}\|_2 \geq \frac{\log(C) - \hat{\mathbb{E}}_{\mathbf{x}\sim\mathcal{D}}[\mathsf{H}[p(\cdot|\mathbf{x};\boldsymbol{\theta})]]}{\sqrt{\boldsymbol{\nu}(\Phi, p_{\mathsf{x}})} - \sqrt{\frac{2}{N}\left(\boldsymbol{\nu}(\Phi, p_{\mathsf{x}}) + \sqrt{\frac{\mathsf{Var}_{p_{\mathsf{x}}}[\|\Phi(\mathbf{x})\|_2^2](2/\delta - 1)}{N}}\right)\log(\frac{2}{\delta})}} \tag{55}$$

$$\|\mathbf{w}\|_2 \geq \frac{\log(C) - \hat{\mathbb{E}}_{\mathbf{x}\sim\mathcal{D}}[\mathsf{H}[p(\cdot|\mathbf{x};\boldsymbol{\theta})]]}{\left(2 - \sqrt{\frac{2}{N}\log(\frac{2}{\delta})}\right)\sqrt{\boldsymbol{\nu}(\Phi, p_{\mathsf{x}})} - \boldsymbol{\Theta}\left(N^{-0.75}\right)} \tag{56}$$

$\square$

## Appendix 3: Training Details on FGVC

**ResNet-50:** Training is done for 40k iterations with batch-size 8 with an initial learning rate of 0.005. Optimal $\gamma$ for each dataset is given in Table 1.

| Dataset | $\gamma$ |
|---|---|
| CUB2011 | 0.9 |
| NABirds | 0.7 |
| Stanford Dogs | 0.7 |
| Cars | 0.8 |
| Aircraft | 1 |

Table 1: Regularization parameter $\gamma$ for ResNet-50 experiments.

**Bilinear and Compact Bilinear CNN:** We follow the training routine given by the authors[1]. Optimal $\gamma$ for each dataset is given in Table 2.

| Dataset | $\gamma$ |
|---|---|
| CUB2011 | 1 |
| NABirds | 1 |
| Stanford Dogs | 1 |
| Cars | 1 |
| Aircraft | 1 |

Table 2: Regularization parameter $\gamma$ for Bilinear CNN experiments.

**DenseNet-161:** Training is done for 40k iterations with batch-size 32 with an initial learning rate of 0.005. Optimal $\gamma$ for each dataset is given in Table3.

**GoogLeNet:** Training is done for 300k iterations with batch-size 32, with a step size of 30000, decreasing it by a ratio of 0.96 every epoch. Optimal hyperparameters are given in Table 4.

**VGGNet-16:** Training is done for 40k iterations with batch-size 32, with a linear decay of the learning rate from an initial value of 0.1. Optimal $\gamma$ is given in Table 5.

| Dataset | $\gamma$ |
|---|---|
| CUB2011 | 0.8 |
| NABirds | 1 |
| Stanford Dogs | 0.8 |
| Cars | 1 |
| Aircraft | 0.8 |

Table 3: Regularization parameter $\gamma$ for DenseNet-161 experiments.

| Dataset | $\gamma$ |
|---|---|
| CUB-200-2011 | 10 |
| NABirds | 1 |
| Stanford Dogs | 1 |
| Cars | 1 |
| Aircraft | 1 |

Table 4: Regularization parameter $\gamma$ for GoogLeNet experiments.

| Dataset | $\gamma$ |
|---|---|
| CUB2011 | 1 |
| NABirds | 1 |
| Stanford Dogs | 1 |
| Cars | 1 |
| Aircraft | 1 |

Table 5: Regularization parameter $\gamma$ for VGGNet-16 experiments.

## Footnotes

[1]`https://github.com/gy20073/compact_bilinear_pooling/tree/master/caffe-20160312/examples/compact_bilinear`

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

Figure 1: We get consistent improvement in validation accuracy as the amount of training data is increased. Curves plotted for various values of $\gamma$ on CIFAR10 with model ResNet20.