[Reviews · NeurIPS 2018]

Reviewer 1



This paper presents a simple and effective approach for fine-grained image recognition. The core idea is to introduce max-entropy into loss function, because regular image classification networks often fail to distinguish semantically close visual classes in the feature space. The formulation is clear and the performance is very good in fine-grained tasks. I like the ablation study on CIFAR10/100 and different subsets of ImageNet, showing that this idea really works in classifying fine-grained concepts. The major drawback of this paper lies in its weak technical contribution. Max-entropy is a well-known theory, and this paper seemed not to provide any improvement on it, but just applied it to fine-grained recognition (I searched on Web, but I did not see any work with the same solution, so I vote for moderate novelty). It would be a good contribution to the computer vision community, but not a significant one to NIPS which leans more to theory. In addition, another concern is that the formulation is closely similar to that in [30] (see the first equation in Sec 3 (arXiv: 1701.06548), KL divergence and cross-entropy differ from each other by a constant). Please clarify the difference, otherwise this work would be considered much less novel. ===== POST-REBUTTAL COMMENTS ===== After reading the rebuttal and other reviewers' comments, I can accept that this paper is an extension to [30] with a proof why a max-entropy-based approach works well on fine-grained recognition. Again, this is a nice work, but the technical contribution is not significant, which limits its quality especially for NIPS. Considering other reviewers' comments, I would change my rating to weak accept.

Reviewer 2



The paper is very clear, provides a good mathematical description, is pleasant to read and expresses in clear terms the problem it is trying to solve, namely improving fine-tuning performances for fine-grained classification using maximum entropy training. MET is relatively easy to implement and does improve performance. However, on some examples the gain is very small. This is an incremental contribution but one of good quality.

Reviewer 3



This paper proposes a fine-grained image recognition method with maximum-entropy learning. The proposed method is easy-to-implement and simple to understand, which brings scalability and generalizations. Some details: + This paper is well written and clear. + The proposed method could bring consistent improvements on different base models on fine-grained image recognition. Meanwhile, it also conducts experiments on general image recognition datasets. + The theoretical part is sufficient. Minor issues: - The format of references is not consistent. - In the title of Fig. 1, CUB-200-2011 could be wrong, which should be Stanford Dogs.